# Investigations into the Recognisability of Gear Damage Sizes in Vibration Signals and Calculation of Appropriate Digital Filter Limits

Andreas Beering *[ID] and Karl-Ludwig Krieger [ID]

Institute of Electrodynamics and Microelectronics, University of Bremen, 28359 Bremen, Germany; krieger@item.uni-bremen.de
* Correspondence: beering@item.uni-bremen.de

**Abstract:** The present work investigates the size of gear damage required for significant recognisable change in the vibration signal and presents a method to determine digital filter limits in order to emphasise the vibration behaviour in the time domain. For this purpose, two gears are artificially damaged to four different degrees. The damage levels are determined by a tactile gear measurement and the gears are inserted into two intact gearboxes. Measurements at different speeds are used to generate a representative dataset. On the one hand, the recorded signals are examined via cross-correlation in the time domain. On the other hand, the occurring frequency components are examined using a windowed fast Fourier transformation. Based on the two observations, a statement is made about the recognisability of the damage levels of the two gears in the vibration signal. Furthermore, smoothed spectra are calculated via linear prediction coefficients (LPC) and an appropriate number of required coefficients is estimated via the Akaike information criterion. Subsequently, the calculated prediction coefficients are used as coefficients of an all-pole filter to calculate difference spectra. Based on the difference spectra, filter limits for a digital filter are derived to emphasise the damaged tooth meshing in the time domain.

**Keywords:** digital filter limits; gear damage; linear prediction; signal processing; vibration analysis





## 1. Introduction

Condition monitoring is becoming increasingly important in many areas, such as gearbox or drive monitoring. In most cases, condition monitoring aims at the intelligent evaluation of sensor data for the derivation of a machine's condition and the estimation of its remaining useful lifetime. This is based on sensor values that are recorded on machine-relevant components. Since sensors only detect changes above a certain size, this paper first investigates the size of damage to tooth flanks at which changes occur in the recorded vibration signals. In addition to monitoring vibrations, there are many other influencing factors, such as lubricants or machine currents, which are investigated in the area of condition monitoring. A comprehensive overview of other methods is given in [1]. The focus of this approach is on the analysis of vibration signals. For this purpose, gearboxes with synthetically damaged tooth flanks are examined more closely in the time and frequency domain. Furthermore, the aim of this paper is to calculate digital filter limits to improve the recognisability of damage in the vibration signal.

Previous approaches and investigations into gear tooth flanks are mainly aimed at the detection of damage. In [2], synthetic damage to gears is examined. For this purpose, entire teeth are removed from the gear and subsequently, the vibration data are recorded and evaluated. A comparable investigation is carried out in [3]. Here, entire teeth were removed and a damage analysis was carried out. The investigation is based on the study of the current of the drive. In [4,5], real tooth fractures are investigated, which are also quite massive. In [6], different degrees of damage to gears are investigated. Here, the examined

teeth of the gears are removed by 25%, 50% and 100% from the top. In [7], two methods for modelling vibrations are presented and compared with experimental investigations. The tooth damage studied is quite small, but not precisely quantified. In [8], we presented an analysis of the tooth damage of gears under real load conditions, in which time-domain data are investigated via a correlation-based approach. The studies shown have in common that the damage is very massive or not precisely quantified. Furthermore, it is unclear from which damage level onwards that the damage becomes recognisable in the vibration signal. Investigations such as [9,10] into the wear and remaining useful lifetime of gearboxes usually do not deal with this aspect either, as these investigations examine the long-term changes of previously determined characteristics. In contrast to the previous studies, this paper presents a novel method for calculating the digital filter limits of gear damage. Furthermore, this paper demonstrates, for the examined gearbox type, the measurable damage level to tooth flanks from which a change in the vibration signal occurs.

This paper first presents an investigation of the different degrees of damage on two different gears. A metric of gear measurement is used to quantify the different damage levels. In the analysis of the measurements, the data sets are first segmented so that a statistically representative number of damaged tooth meshes can be considered for each damage level. Subsequently, a correlation-based investigation in the time domain and a comparison of the frequency spectra of damaged tooth meshes are described in order to make a statement on the recognisability of quantified damage levels in the vibration signal. In order to derive appropriate filter limits, smoothed spectra are first calculated via linear prediction coefficients [11] and then filter limits are determined via difference spectra. Here, a number of filter coefficients are estimated via the Akaike information criterion.

This contribution is structured as follows. Section two describes the experimental setup. First, the test rig for operating the gearbox is presented. This is followed by a more detailed description of the structure of the gearbox under investigation, the synthetic damage levels and their theoretical occurrence in the vibration signal. The third section deals with the analysis of the measurement series of the damaged gears. First, the data are segmented and then analysed in the time and frequency domain in order to make a statement about the recognisability of the damage in the vibration signal. Section four deals with the theoretical background of linear prediction and the estimation of an appropriate order of prediction coefficients via the Akaike information criterion. Furthermore, smoothed spectra of the vibration signals are calculated using these coefficients. Subsequently, in section five, filter limits for digital filters are determined via the difference spectra of the previously smoothed spectra in order to be able to emphasise the damage in the time signal via digital filtering. Finally, a conclusion of the contribution is given.

## 2. Experimental Setup

This section describes the experimental setup, the gear examined and the synthetic damage levels. First, the test rig, which is used for the vibration measurements on the gearboxes, is presented. Afterwards, the structure of the examined gearbox and the synthetic damage types are described. Finally, the theoretical occurrence of damage in the vibration signal is described.

### 2.1. Test Rig

Figure 1 shows a schematic drawing of the test rig for the laboratory investigation of gearbox tooth damage. On the right side of the figure there is an oil engine, which drives the gearboxes connected to it at the input shaft with an adjustable rotational speed. A shaft is coupled to the output of the gearbox, driving a second gearbox of the same type. The second gearbox is used, together with the following pump, to apply a load torque to the drive for the gearbox under investigation. The second gearbox drives the pump, which pumps oil through a closed oil circuit. An adjustable pressure relief valve limits the flow of oil through the circuit and allows the load torque to be varied. The pressure relief valve is electrically controllable. An oil tank is used to minimise temperature influences

on the oil, which can result in a change in torque. A torque-measuring shaft is inserted between the input shaft of the second gearbox and the pump to measure both the torque and the rotational speed of the input shaft of the gearbox. Since both gearboxes have the same gear ratio, the measured rotational speed and torque also correspond to that of the gearbox under investigation. To measure the occurring gearbox vibrations, a piezoelectric vibration transducer is attached to the gearbox housing (material: cast iron) via a screw connection in order to ensure good material coupling of the sensor [12]. A iCS80 [13] sensor was used. It has a linear frequency range from 0.13 Hz to 22 kHz (3 dB cut-off frequency), a measuring range of $\pm 55$ g and a voltage sensitivity of 100 mV/g. The signal of the piezoelectric vibration sensor is sampled at a sampling rate of $f_s = 51.2$ kHz and a resolution of 24 bit. The signal is resampled to a frequency of 44 kHz. This still allows the full sensor range to be examined and results in a better dynamic range, as the quantisation noise is distributed over the entire sampled signal bandwidth and is therefore lower in the useful signal bandwidth after the resampling has been applied [14].

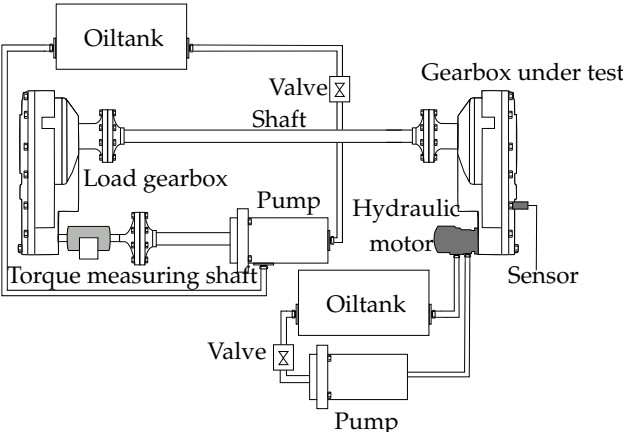

**Figure 1.** Structure of the test rig for the investigation of gearboxes and the mechanical coupling of the piezoelectric vibration transducer.

*2.2. Schematic Gearbox Structure*

Figure 2 shows the schematic structure of the gearbox under investigation. The gearbox under study is composed of a total of four gears. All gears are made of steel and have a pressure angle of $20°$, a helix angle of $0°$ and no modifications. The four gears $z_1$, $z_2$, $z_3$ and $z_4$ have $n_{z_1} = 14$, $n_{z_2} = 50$, $n_{z_3} = 11$ and $n_{z_4} = 47$ teeth.

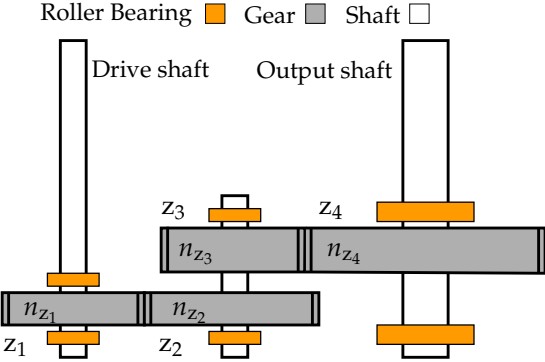

**Figure 2.** Schematic structure of the examined gearbox.

Further parameters of a gear measurement of the examined gear type are listed for reference in Table 1.

**Table 1.** Parameters of gear measurement of an intact gearbox.

|  | $z_1$ | $z_2$ | $z_3$ | $z_4$ |
|---|---|---|---|---|
| Module [mm] | 3 | 3 | 4 | 4 |
| Total profile deviation [µm] | 47.3 | 48.6 | 41.3 | 36.6 |
| Single pitch deviation [µm] | 20.3 | 18.3 | 14.0 | 27.6 |
| Adjacent pitch deviation [µm] | 35.3 | 31.1 | 27.7 | 39.8 |
| Total pitch deviation [µm] | 28.7 | 56.4 | 19.9 | 74.8 |
| Accuracy grade (ISO1328-1 [15]) | 11 | 9 | 9 | 11 |

Due to the bad accuracy grades, higher vibration levels are to be expected. The gearbox is composed of two gear stages. The gear ratios $i_1$ and $i_2$ of the individual gear stages result from the number of meshing teeth $n_{z_2}/n_{z_1}$ of the gears $z_2$ and $z_1$, respectively, and $n_{z_4}/n_{z_3}$ of the gears $z_4$ and $z_3$, shown in Figure 2 according to

$$i_1 = \frac{n_{z_2}}{n_{z_1}} = \frac{50}{14} \approx 3.57 \tag{1}$$

$$\text{and } i_2 = \frac{n_{z_4}}{n_{z_3}} = \frac{47}{11} \approx 4.27. \tag{2}$$

The total gear ratio $i_{12}$ of the gearbox is obtained by multiplying the individual gear ratios; $i_{12} = i_1 \cdot i_2 \approx 15.26$.

### 2.3. Synthetic Gear Damage

For the investigations into the recognisability of damage levels, four teeth of each two gears were artificially damaged. Gears $z_1$ (14 teeth) and $z_3$ (11 teeth) of the gearbox were examined. Four teeth of the two gears were damaged with synthetic flank damage. For gear $z_1$, teeth 1, 4, 7 and 10 and for gear $z_3$, teeth 1, 3, 6, 9 were damaged. This synthetic flank damage was realised by removing parts of the tooth flanks. Here, the damage size was iteratively reduced so that the transition from recognisable to non-recognisable damage could be examined through the damage levels. Each of the gears were inserted into a new gearbox to ensure that only the influence of the specific gear occurs in the vibration signal. The respective gears were operated with the test rig from Figure 1. The single pitch deviation was used as a way of measuring the magnitude of the synthetic damage levels. Single pitch deviation is a common metric for gear measurement according to DIN 3960 [16]. The single pitch deviation $f_P$ is shown in Figure 3.

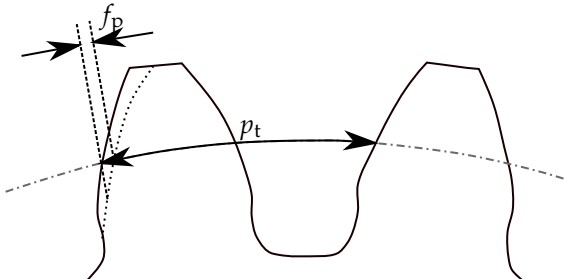

**Figure 3.** Circular pitch $p_t$ and single pitch deviation $f_P$ of a gear according to [17].

The circular pitch of a gear results from the pitch diameter $d_t$ and the number of teeth $n_z$ according to $p_t = \frac{\pi d_t}{n_z}$. The single pitch deviation $f_P$ describes the difference between the circular pitch $p_t$ and the real pitch [18]. The single pitch deviation of the measured gears is given in µm. The values of the four teeth of each gear were determined via a tactile gear measurement. Table 2 shows a listing of the measured deviations for both the synthetic damaged gears. In addition to the four damage levels $S_1$–$S_4$, the average deviation of an intact gear $S_0$ of the same type without damages is given as reference.

**Table 2.** Single pitch deviation $f_p$ of the two examined gears $z_1$ and $z_3$.

| | Single Pitch Deviation $f_p$ [µm] | | | | |
|---|---|---|---|---|---|
| | $S_0$ | $S_1$ | $S_2$ | $S_3$ | $S_4$ |
| $z_1$ | 13.40 | 344.02 | 73.48 | 52.75 | 19.35 |
| $z_3$ | 14.00 | 1504.31 | 847.40 | 263.18 | 65.94 |

Since gears have a periodic meshing behaviour, signal changes in the vibration signal caused by damage also occur periodically.

Figure 4a shows the theoretical occurrence of damage for gear $z_1$ with 14 teeth (damage to teeth 1, 4, 7 and 10) and Figure 4b shows that for gear $z_3$ with 11 teeth (damage to teeth 1, 3, 6 and 9). The period $T_{z_i}$ in which the gear has rotated once results from the geometry of the gear and the rotational speed of the input shaft $\omega(t)$. For gear $z_1$, the period in seconds is given by $T_{z_1} = 60/\omega(t)$. For gear $z_3$, the calculation for the period $T_{z_3}$ additionally contains multiplication with the gear ratio $i_1$, since the gear is on the second gearbox stage. Since the frequency of occurrence in the damaged tooth meshes depends significantly on the rotational speed, different speed levels were examined for each gearbox as part of these investigations. Figure 5 shows the speeds used and the theoretical occurrence of the damage.

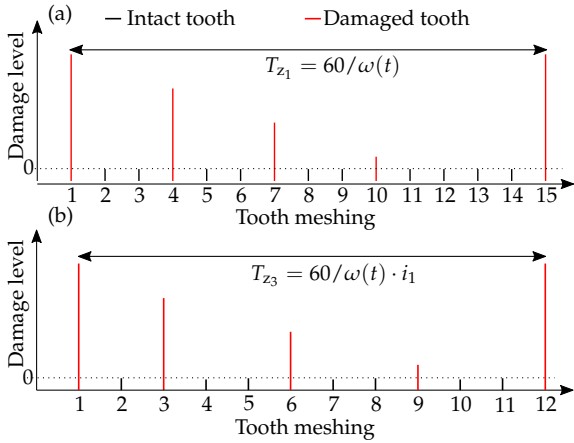

**Figure 4.** Theoretical occurrence of tooth meshing of damaged teeth for gear $z_1$ in (**a**) and $z_3$ in (**b**).

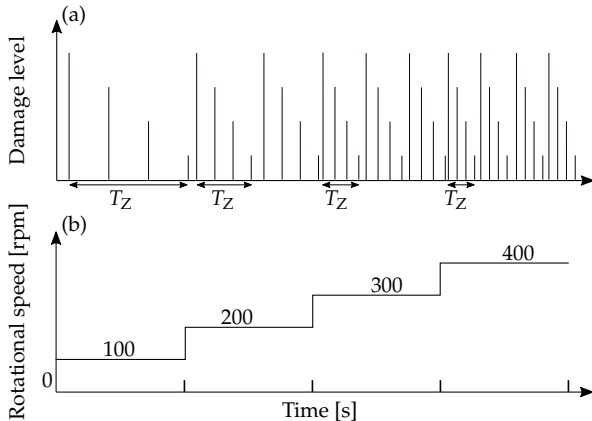

**Figure 5.** Theoretical occurrence of tooth meshing of damaged teeth (**a**) for different rotational speeds (**b**).

The gearbox is used to drive a chain transport floor. Preliminary investigations have shown that the rotational speed is in a range between 100 and 400 rpm. In addition, it has been shown that the rotational speed changes only very slowly during operation. Based on

these findings, four input shaft speeds of approximately 100, 200, 300 and 400 rpm were selected for these investigations. The torque used during the measurement varied between 20 and 42 Nm at the input shaft of the gearbox. Since the speed values in the design are controlled by the oil flow, deviations in the rotational speed due to temperature and thus viscosity can occur in the investigations. These are usually very small and do not affect the analysis.

## 3. Data Analysis

This section describes the analysis of the measurement series. First, the segmentation of the data sets is presented in order to create comparable signals. Subsequently, the segmented signals in the time domain are compared with regard to the recognisability of the damage. Finally, the signals are examined in the frequency domain.

### 3.1. Data Segmentation

In the following, the vibration signal of one gear revolution at a rotational speed approximately of 200 rpm is considered as an example for each gear. For this purpose, the recorded vibration signal is transformed into the time–frequency domain as a spectrogram. Spectrograms of vibration signals are determined in digital signal processing via the discrete short-time Fourier transform (STFT). The discrete STFT $\mathcal{F}_{x,m,k}^{\gamma}$ [19] is given by

$$\mathcal{F}_{x,m,k}^{\gamma} = \sum_{n=0}^{N-1} x[n] \cdot \gamma^*[n - m\Delta M] \cdot e^{\frac{-j2\pi kn}{N}}. \tag{3}$$

Here, $x[n]$ describes a discrete-time signal and $\gamma^*[n - m\Delta M] \cdot e^{\frac{-j2\pi kn}{N}}$ a time- and frequency-shifted window function in the considered interval $[0, N-1]$. Since only discrete frequencies and time points are considered, $m = 0, 1, \ldots, M-1$ applies. $\Delta M$ describes the considered window size. The complex-valued short-time Fourier transform is converted into real numbers via the magnitude square for the pictorial representation in a spectrogram $\mathcal{S}_{x,m,k}$:

$$\mathcal{S}_{x,m,k} = \left| \mathcal{F}_{x,m,k}^{\gamma} \right|^2 = \left| \sum_{n=0}^{N-1} x[n] \cdot \gamma^*[n - m\Delta M] \cdot e^{\frac{-j2\pi kn}{N}} \right|^2 \tag{4}$$

The spectrogram in decibels $\mathcal{S}_{x,m,k,\text{dB}}$ results in $\mathcal{S}_{x,m,k,\text{dB}} = 20 \cdot \log_{10}(\mathcal{S}_{x,m,k})$. First, in Figure 6, the vibration signal is considered above and the corresponding spectrogram for the measurement of the gearbox with synthetic damage to gear $z_1$ is considered below.

The vibration signal shown was sampled at 51.2 kHz. A resampling to 44 kHz was applied. For the calculation of the following spectrograms, an FFT length of 512 values, a Blackman window function with a length of 512 values and an overlap of 511 values were chosen. Due to the short FFT length and the large overlap, a good temporal resolution can be achieved, which is necessary for the assignment of the individual teeth. Furthermore, the black lines show the parts of the signal in which the tooth meshes of the synthetically damaged teeth occur. These time windows are labelled according to the damaged teeth $S_1, S_2, S_3, S_4$ from Section 2.3. The highlighted signal parts were determined based on the spectrogram, the rotational speed and the known geometry of the synthetic damage. The highlighted areas correspond to the proportionate duration of the respective teeth meshes. The largest damage ($S_1$) was determined in the spectrogram via its period duration and subsequently the positions of $S_2, S_3, S_4$ were calculated. The procedure for selecting the time windows will be discussed in more detail later in this section.

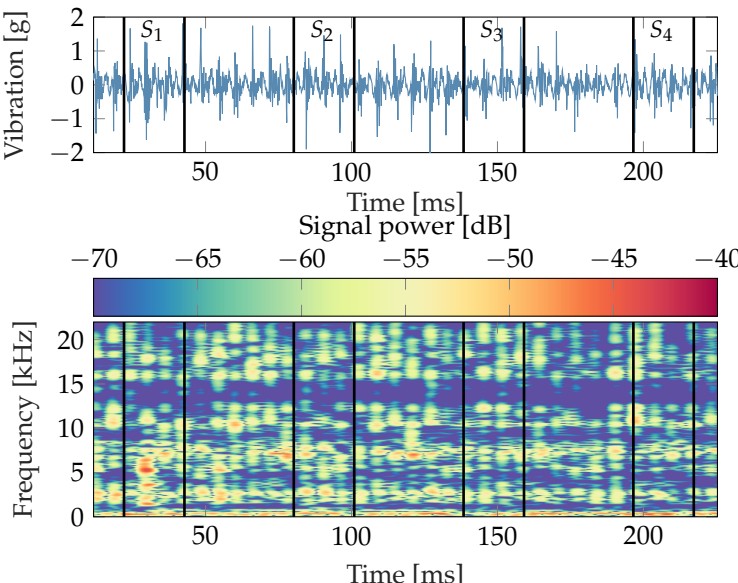

**Figure 6.** Vibration signal and spectrogram of the artificially damaged gear $z_1$.

The signal in the time domain shows the vibrations over a time interval of about 200 ms. The vibration signal is composed of many individual transients, which are caused by the metallic contacts of the tooth meshes and roller bearings. In $S_1$, a more distinctive transient can be seen compared to the rest of the signal. In terms of amplitude, however, this can hardly be distinguished from the other transients. For $S_2$, $S_3$ and $S_4$, no differences to the undamaged tooth meshes can be seen in the time domain. In the spectrogram of the vibration signal, the tooth mesh of the synthetic damaged tooth in $S_1$ can be clearly recognised by the more distinctive frequency components in a range from 1 kHz to 7 kHz. This transient can be clearly separated from the entire signal shown. $S_2$ shows smaller distinct frequency ranges which cannot be clearly assigned to the damage due to its comparable occurrence in the intact tooth meshes. For $S_3$ and $S_4$, no noticeable distinctive frequencies occur.

An analogous consideration for the gearbox with artificially damaged gear $z_3$ is shown in Figure 7.

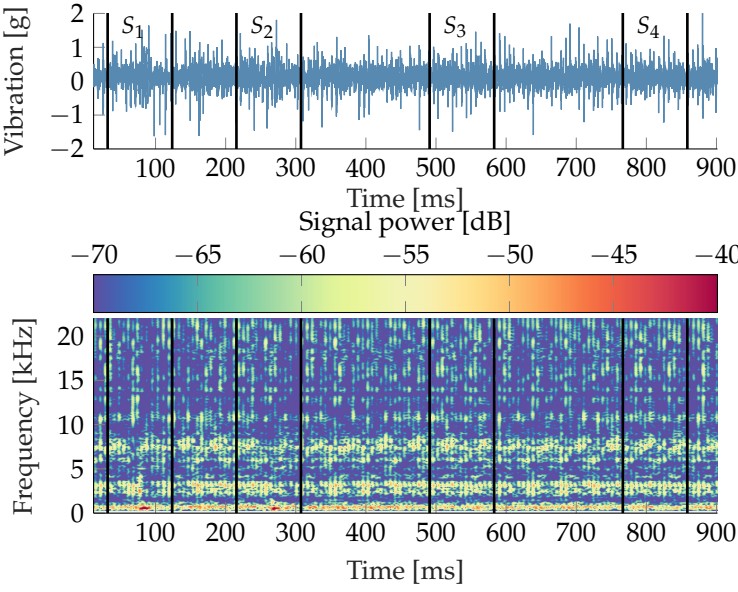

**Figure 7.** Vibration signal and spectrogram of the artificially damaged gear $z_3$.

Here again, one gear revolution is considered. Compared to the previous observation, this results in a length of about 900 ms due to the gear ratio $i_1 = 3.57$. In this observation, no concrete deviation from the rest of the signal can be recognised in the time domain even in the time window $S_1$ of the greatest damage. Only when looking at the time–frequency domain in the bottom illustration, can the differences be recognised. Above all, $S_1$ is characterised by strongly distinctive frequencies in the range of about 1 kHz. This distinctive frequency range also occurs for $S_2$. $S_3$ and $S_4$ show no differences to the intact tooth meshes.

For a statistical evaluation of the tooth meshing of the synthetically damaged teeth, the individual measurements are first segmented. Based on the previous observation, it could be shown that for the two examined gears, the most severe damage $S_1$ is recognisable in each case by a more distinctive frequency behaviour in the spectrogram. In order to first determine the times of these damaged tooth meshes, the mean signal power of the spectrogram $\bar{s}_{x\Delta f,m,dB}$ in the dominant frequency ranges $[f_1, f_2]$, which were previously derived, is calculated according to

$$\bar{s}_{x\Delta f,m,dB} = 20 \cdot \log_{10}\left(\frac{\sum_{k=f_1}^{f_2} \mathcal{S}_{x,m,k}}{(f_2 - f_1)}\right). \tag{5}$$

As an example for the segmentation of the tooth meshes of the damage $S_1$, a spectrogram as well as the mean signal power $\bar{s}_{x\Delta f,m,dB}$ in the range of 1 kHz to 7 kHz is shown in Figure 8. The peaks in the bottom plot correspond to the damaged tooth meshes. In addition, it can be seen that there are no anomalies in the signals in the areas after the damage, and thus the intact meshes are not influenced by the damaged ones before them. The vibration signal used for this spectrogram was recorded at the gearbox with damage to gear $z_1$. The measured speed is initially constant at 207 rpm and increases to 322 rpm after about one second.

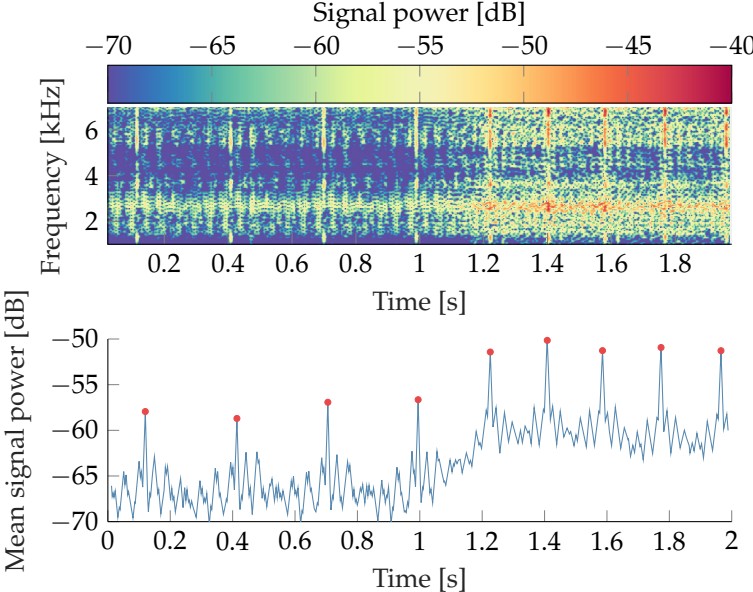

**Figure 8.** Spectrogram (**top**) and mean signal power (**bottom**) of the vibration signal of the gearbox with synthetic damage at gear $z_1$ in the range of 1–7 kHz.

In the spectrogram, recurring significant distinct frequencies can be recognised. Furthermore, an increase in the total signal power can be seen from about one second onwards, due to the increase in rotational speed at this point. The average calculated signal power

in the range of 1–7 kHz also shows the signal peaks. The total average power increase due to the increased speed can also be seen in this diagram. Furthermore, it can be seen that the period duration decreases due to the increasing speed. The average period $\bar{T}_{z_1}$ between the determined signal peaks in the range of the measured speed of $\omega[t] = 207$ rpm is $\bar{T}_{z_1} = 0.2915$ s. The theoretical period duration $\bar{T}_{z_1,\text{theo}}$ for the occurrence of the damage at $z_1$ is given by

$$\bar{T}_{z_1,\text{theo}} = \frac{1}{\omega[t]/60} = \frac{1}{207\,\text{rpm}/60} = 0.2899\,\text{s} \tag{6}$$

and thus coincides within the scope of the measurement accuracy with the period durations determined via the average signal power. The mean period between the signal peaks in the speed range of 322 rpm is 0.1849 s. This again coincides with the theoretical period duration of 0.1863 s for this speed. As a result, the signal peaks can be assigned to the damage $S_1$. The signal peaks are defined as the times $t_i(S_1)$ of the tooth meshes $S_1$. Based on the geometry and the known number of teeth being 14, the times $t_i(S_2)$, $t_i(S_3)$ and $t_i(S_4)$ of the damaged tooth meshes $S_2$, $S_3$ and $S_4$ are given by

$$t_i(S_2) = t_i(S_1) + 3 \cdot \frac{\bar{T}_{z_1}}{14}, \tag{7}$$

$$t_i(S_3) = t_i(S_1) + 6 \cdot \frac{\bar{T}_{z_1}}{14}, \tag{8}$$

$$\text{and } t_i(S_4) = t_i(S_1) + 9 \cdot \frac{\bar{T}_{z_1}}{14}. \tag{9}$$

In addition, the time $t_i(S_0)$ of intact tooth meshes was determined as a reference for the following consideration according to

$$t_i(S_0) = t_i(S_1) + 11 \cdot \frac{\bar{T}_{z_1}}{14}. \tag{10}$$

The length of the respective signals is $\bar{T}_{z_1}/14$, as this corresponds to the part of the respective tooth during one gear revolution. The signals of the tooth meshes are segmented based on these times and the interval length $\bar{T}_{z_1}/14$. An analogous consideration for determining the signals was carried out for the measurement series of the gearbox with damage to gear $z_3$. The total number of signals $n_{S,z_1}$ recorded for each of the four damages on gear $z_1$ and the number $n_{S,z_3}$ for damages on gear $z_3$ for the respective rotational speeds can be taken from Table 3.

**Table 3.** Number of signals $n_{S,z_1}$ for each of the four damages on gear $z_1$ and $n_{S,z_3}$ for each of the four damages on $z_3$ for the respective rotational speeds.

| | Rotational Speed [rpm] | | | |
|---|---|---|---|---|
| | **100** | **200** | **300** | **400** |
| $n_{S,z_1}$ | 13 | 29 | 41 | 50 |
| $n_{S,z_3}$ | 14 | 34 | 48 | 64 |

For each rotational speed, a statistically representative set of signals was thus determined for each damage level.

### 3.2. Recognisability of Gear Damage in the Time Domain

In order to examine the experimentally possible recognisability of the synthetic damages, the segmented signals are examined more closely in the time domain. For this purpose, the cross-correlation between the individual signals determined beforehand is calculated in order to use the correlation coefficients to make statements about the damage levels and the recognisability of damage. The cross-correlation function (CCF) is a modification of an

autocorrelation function (ACF). A discrete autocorrelation $R_{x_1 x_1, m}$ describes the correlation of a signal $x_1[n]$ with itself at another time $x_1[n + m]$ over a considered interval $[-M, M]$ and is described by

$$R_{x_1 x_1, m} = \lim_{M \to \infty} \frac{1}{2M + 1} \sum_{n = -M}^{M} x_1[n] \cdot x_1[n + m].$$  (11)

The autocorrelation function thus describes the similarity of a signal part to previous times in the signal [20]. The cross correlation is an extension of the ACF in which a signal $x_1[n]$ is compared with a second different signal $x_2[n]$ [20]. For the discrete case, the cross correlation $R_{x_1 x_2, m}$ between two signals is calculated according to

$$R_{x_1 x_2, m} = \lim_{M \to \infty} \frac{1}{2M + 1} \sum_{n = -M}^{M} x_1[n] \cdot x_2[n + m].$$  (12)

The cross correlation describes the similarity of two signals $x_1[n]$ and $x_2[n]$. The cross correlation is normalised over the values of the ACF without shift $R_{x_1 x_1, 0}$ and $R_{x_2 x_2, 0}$ of both signals to

$$R_{x_1 x_2, \text{coeff}, m} = \frac{1}{\sqrt{R_{x_1 x_1, 0} R_{x_2 x_2, 0}}} R_{x_1 x_2, m}.$$  (13)

This results in values in the range $[-1, 1]$ for the normalised CCF, which are thus comparable to the values of the correlation coefficients of the Pearson correlation [21]. Since it cannot be assumed that all signals recorded are ideally aligned, the maximum value of the CCF is calculated. Signals can be aligned via the associated temporal offset of the maximum value [22]. The maximum value of the cross correlation function $r_{\max}$ thus describes the similarity of two signals aligned to each other and results in

$$r_{\max} = \max \left( R_{x_1 x_2, \text{coeff}, m} \right).$$  (14)

In the following, a correlation matrix of the recorded vibration signals of the gearbox with damage to gear $z_1$ at a rotational speed of 200 rpm is first considered as an example. For this purpose, 34 signals are considered for each damage level $S_1$, $S_2$, $S_3$ and $S_4$, as well as for the reference tooth mesh of the intact teeth $S_0$. Figure 9 shows the correlation matrix for all 170 signals considered. This allows for the different vibration behaviour of the individual damage levels ($S_1 - S_4$) of the gear to be compared with the intact tooth mesh $S_0$. Based on the periodic behaviour of the tooth meshes, a statistically representative set of 34 meshes of each damage level as well as the intact mesh were recorded.The individual damage levels are summarised as blocks and labelled on the y- and x-axis in the figure.

The main diagonal of the matrix shows the correlation of the respective signal with itself. Accordingly, the main diagonal shows correlation coefficients of 1. Furthermore, the blocks on the main diagonal also show increased values, due to the similarity of the individual signals of a damage level. The other blocks thus show the similarity between the individual damage levels. Noticeably, the block from $S_1$ to $S_1$ shows a very high overall correlation. This can be explained by the more distinctive, recurring vibration signal that occurs due to the damage. Furthermore, the blocks between $S_3$ and $S_4$ to $S_1$ show very low correlations overall and thus little similarity to the vibration signal of the damaged meshing. A slightly increased similarity occurs between $S_2$ and $S_1$. In order to make a better statement about the behaviour of the individual damage classes in relation to each other, the correlation coefficients of the individual blocks are averaged. Furthermore, these correlation matrices are determined for all investigated rotational speeds. These matrices are also averaged. Figure 10 shows the correlation matrices of the measurements of the gearbox with damage at gear $z_1$ (left) and gear $z_3$ (right) averaged over the speeds. In addition, the mean correlation coefficients are shown in the blocks.

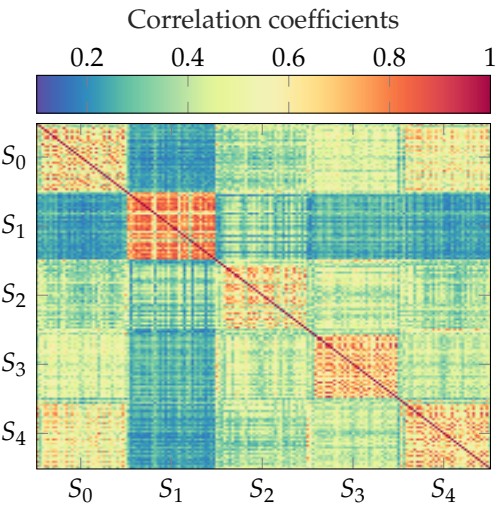

**Figure 9.** Maximum correlation coefficients of the cross-correlation of all signals for the gearbox with damage at gear $z_1$.

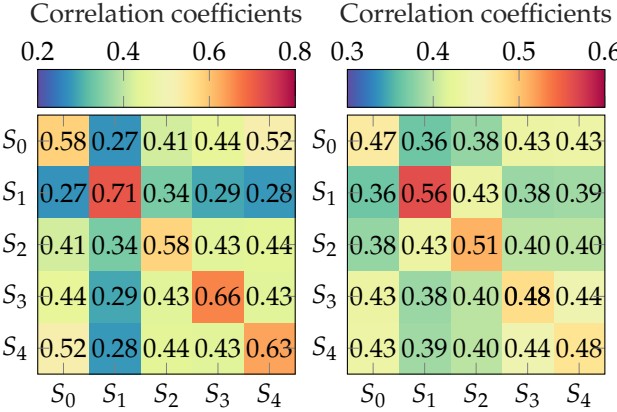

**Figure 10.** Maximum correlation coefficients of the cross-correlation for the gearbox with damage to gear $z_1$ (**left**) and for the gearbox with damange to $z_3$ (**right**) averaged over equal damage sizes.

The main diagonals of both correlation matrices show the highest values of the respective damage levels in this observation. This can be explained by the high similarity of the periodically occurring gear meshing. Overall, the mean correlation coefficients of the gearbox with damage to gear $z_1$ show significantly higher correlation coefficients than those of the gearbox with damage to gear $z_3$, due to the longer signal length caused by the slower rotational speed. For the correlation matrix of the measurement series on the gearbox with damage to $z_1$, it can be seen that the damage level $S_1$ clearly distinguishes itself from the others. It shows an increased correlation of 0.71 to the signals among each other. Furthermore, $S_1$ clearly differs from $S_0$, $S_3$ and $S_4$. The slight similarity of $S_2$ to $S_1$ indicates that the damage level $S_2$ also exhibits recognisable characteristics. The damages $S_3$ and $S_4$ show the highest similarity to the reference signal $S_0$. From this it is concluded that these two signals do not show clear distinguishability and thus the damage level cannot be detected. The correlation matrix of the measurement series on the gearbox with damage to $z_3$ shows a comparable pattern. Here, especially damages $S_1$ and $S_2$, stand out from the other damages and show an increased correlation to each other. Furthermore, damage $S_1$ shows the highest correlation among the signals, due to the distinctive signal behaviour already recognised in the spectrograms. Since damages $S_3$ and $S_4$ show the greatest similarity to the reference signals $S_0$, no distinguishability of the damages appears to be possible.

Overall, based on this investigation, damages $S_1$ and $S_2$ show a recognisable behaviour in the time domain for both gears, since they stand out due to their increased correlation

to each other and also due to their lower correlation to damages $S_3$ and $S_4$, as well as the signals of the intact tooth meshing $S_0$.

### 3.3. Recognisability of Gear Damage in the Frequency Domain

In this section, the previously segmented signals will be compared in the frequency domain in order to make statements about the recognisability of the damages via a further distinguishing feature. For the investigations in the frequency domain, the segmented signals are transformed into the frequency domain. For this, the discrete Fourier transformation $X_k$ of a discrete signal $x[n]$ is used. This was already implicitly used in the calculation of the STFT in Section 2.2. It results in

$$X_k = \sum_{n=0}^{N-1} x[n] e^{\frac{-j2\pi kn}{N}} \text{ for } k = 0, 1, \ldots, (N-1). \tag{15}$$

$N$ describes the number of frequencies. A window function is used to avoid spectral leakage. The discrete Fourier transformation (DFT) of the windowed signal $X_k^\gamma$ therefore results in

$$X_k^\gamma = \sum_{n=0}^{N-1} x[n] \gamma[n] e^{\frac{-j2\pi kn}{N}}. \tag{16}$$

$\gamma[n]$ describes the window function. The FFT is used as an implementation of the DFT. To avoid unequal signal lengths in the frequency range, zero padding [23] is used and zeros are added to the vector after the windowing. The information content of the resulting frequency signal is not changed by zero padding, but the spectrum is smoothed by the increased frequency resolution. The new vector length is chosen based on the longest signal (at a rotational speed of 100 rpm) to the next higher power of two. After calculating all spectra $X_{S_i,k}^\gamma$ for each damage level and rotational speed, an averaged spectrum $\overline{X}_{S_i,k}^\gamma$ of the respective damage level is calculated.

In the following, the averaged spectra for the measurement series of the gearbox with damage to the gear $z_1$ and subsequently that for damage to the gear $z_3$ are considered. The considered frequency range up to 22 kHz results from the linear range of the sensor up to this frequency.

Figure 11 shows that all spectra have a more distinctive behaviour in the lower frequency range. Overall, the higher frequencies show lower amplitudes. In the range below 1 kHz and above 13.5 kHz, the spectra hardly show a distinguishable behaviour. The figure shows that the largest damage, $S_1$, results in a particularly significant difference. The frequency range between 1 kHz and 9 kHz shows clearly more distinctive amplitudes. Further, smaller distinctive frequency bands occur at approximately 10 kHz, 11.5 kHz and 13 kHz. For the spectrum of the second largest damage $S_2$, differences in the spectra are also noticeable. The frequency bands around 1.5 kHz and 7 kHz especially show higher amplitudes. The spectra of damages $S_3$ and $S_4$ show no recognisable differences to the reference signal at all.

Figure 12 shows a significantly better resolution compared to the previous observation, due to the longer signal length resulting from the lower speed during the second gearbox stage. Furthermore, a decreasing signal amplitude can be recognised at higher frequencies. When comparing the spectra, a very similar spectra pattern can be observed for frequencies above 5 kHz, which do not show a clear differentiation. The greatest difference can once again be determined for damage level $S_1$. From 0.5 kHz to 2.5 kHz, several distinctive frequency bands appear. Another distinct frequency band occurs at approximately 3.7 kHz. Likewise, more distinctive frequency bands also occur for damage level $S_2$. These are mainly in ranges from 0.5 kHz to 1.5 kHz, 2.1 kHz to 2.5 kHz and at about 3.7 kHz. For damage levels $S_3$ and $S_4$, there are no noticeable differences to the reference signal. Overall, based on this investigation, damage levels $S_1$ and $S_2$ of both gearboxes show a recognisable behaviour in the frequency domain, as they stand out due to their distinctive frequencies.

Damages $S_3$ and $S_4$ of this gearbox do not show any differences to the reference signal. Since the examinations in the time and frequency domain coincide with regard to the recognisable damage levels, it is assumed that only damage levels $S_1$ and $S_2$ are recognisable for both gears. When comparing the values from Table 2, it can be implied that no generally applicable value can be given for the recognisability of damage levels. Different gears therefore show different behaviour in vibration signals with regard to recognisability.

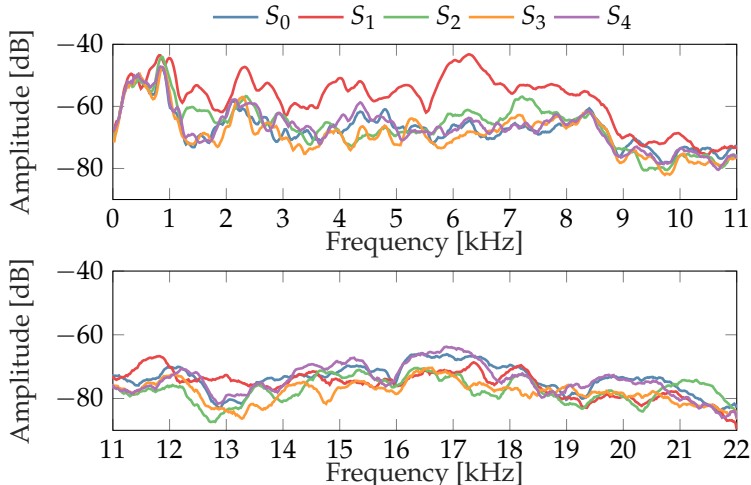

**Figure 11.** Averaged frequency spectra of the measurement series of the gearbox with damage to gear $z_1$.

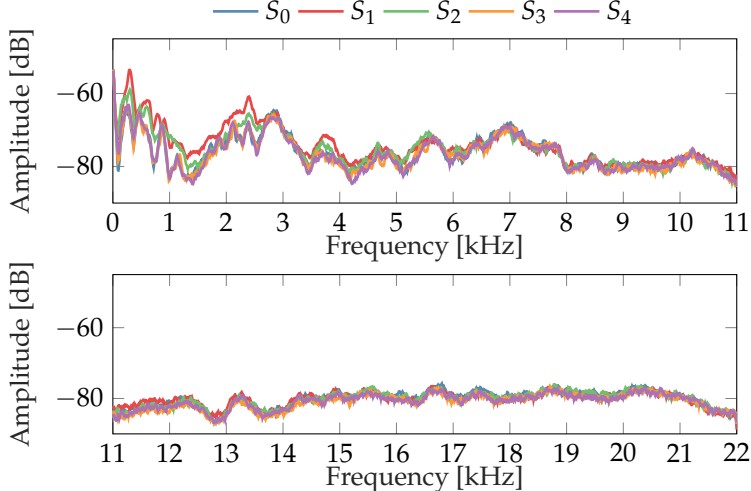

**Figure 12.** Averaged frequency spectra of the measurement series of the gearbox with damage to gear $z_3$.

## 4. Linear Prediction Spectra

This section describes the calculation of difference spectra for the derivation of filter limits. First, the basics of linear prediction and the Akaike information criterion are discussed. Subsequently, the difference spectra are calculated and filter limits are determined.

### 4.1. Linear Prediction Theory

Linear prediction describes a mathematical method of time series analysis in which future values of a signal are estimated based on a linear combination of past values. For this purpose, linear prediction coefficients $\alpha_m$ are calculated according to [24]. These coefficients can be considered as filter coefficients of an all-pole filter. In linear prediction, it is assumed

that a sufficiently stationary signal section $x[n-m]$ is given by a linear prediction $\hat{x}[n]$ according to

$$\hat{x}[n] = -\sum_{m=1}^{p} \alpha_m x[n-m] \tag{17}$$

with the order $p$ of the prediction coefficients. For the calculation of the coefficients, an error term $\varepsilon_{\text{LP},n}$ is further calculated from the input signal and the prediction according to

$$\varepsilon_{\text{LP},n} = x[n] - \hat{x}[n] = x[n] + \sum_{m=1}^{p} \alpha_m x[n-m]. \tag{18}$$

To determine the prediction coefficients, the accumulated squared error $e_{\text{LP}}$ is defined to be

$$e_{\text{LP}} = \sum_{n=-\infty}^{\infty} \varepsilon_{\text{LP},n}^2 = \sum_{n=-\infty}^{\infty} \left( x[n] + \sum_{m=1}^{p} \alpha_m x[n-m] \right)^2. \tag{19}$$

In order to minimise the quadratic error, the following step involves the partial derivation of the error term

$$\frac{\partial e_{\text{LP}}}{\partial \alpha_m} = 0, \quad \text{for } m = 1, \ldots, p. \tag{20}$$

This results in the system of linear equations

$$\sum_{m=1}^{p} \alpha_m \sum_{n=-\infty}^{\infty} x[n-i]x[n-m] = -\sum_{n=-\infty}^{\infty} x[n-i]x[n] \tag{21}$$

with $i = 1, \ldots, p$. By this, $p$ linear equations are given for $p$ unknown parameters $\alpha_1, \ldots, \alpha_p$. The established system of linear equations can be rearranged by the terms of the discrete autocorrelation function $R_{\text{xx}}$ (see Equation (11)). Using the ACF definition and the property that ACFs are even functions ($R_{\text{xx},i} = R_{\text{xx},-i}$), the system of linear equations from Equation (21) can be rearranged to the Yule–Walker equations [25]

$$\sum_{m=1}^{p} R_{\text{xx},|i-m|}\alpha_m = -R_{\text{xx},i}, \quad \text{for } i = 1, \ldots, p. \tag{22}$$

These $p$ linear equations can be described in matrix form as Equation (23).

$$
\begin{bmatrix}
R_{\text{xx},0} & R_{\text{xx},1} & \cdots & R_{\text{xx},p-1} \\
R_{\text{xx},1} & R_{\text{xx},0} & \cdots & R_{\text{xx},p-2} \\
\vdots & \vdots & \ddots & \vdots \\
R_{\text{xx},p-1} & R_{\text{xx},p-2} & \cdots & R_{\text{xx},0}
\end{bmatrix}
\begin{bmatrix}
\alpha_1 \\ \alpha_2 \\ \vdots \\ \alpha_p
\end{bmatrix}
= -\begin{bmatrix}
R_{\text{xx},1} \\ R_{\text{xx},2} \\ \vdots \\ R_{\text{xx},p}
\end{bmatrix}
\tag{23}
$$

The autocorrelation matrix represents a Toeplitz matrix in which the values of the main and secondary diagonals are equal. In digital signal processing, the system of equations is usually solved efficiently via a Levinson–Durbin recursion [25]. As stated before, the estimation of the prediction coefficients $\alpha_m$ of the signal can be interpreted as coefficients of an all-pole filter. This is shown in Figure 13. A frequency spectrum via an FFT together with the spectra via the prediction coefficients is shown. The number of coefficients $p$ was

chosen to be 5, 25, 50, 100 and 1000. For the calculation of the signals, a vibration signal with 8192 samples at a sampling rate of 44 kHz was considered. For better comparability, a frequency range up to 10 kHz is considered.

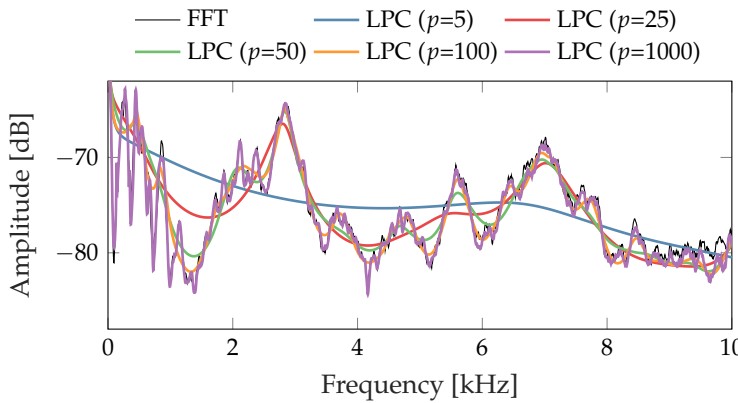

**Figure 13.** Comparison of the spectra via prediction coefficients of different orders $p$ with an FFT spectrum in the frequency range up to 10 kHz.

The effect of the prediction coefficient's order on the spectrum can be seen in the figure. For a very low number of coefficients, the spectrum is strongly smoothed, but hardly resembles the FFT spectrum. If, on the other hand, a very high ordinal number is selected, the spectrum approaches the FFT spectrum.

### 4.2. Order Estimation Using the Akaike Information Criterion

In order to select a reasonable number of coefficients, the Akaike information criterion (AIC) [26] is used. The AIC is used to compare different model parameters in order to evaluate their information content. According to [27], the AIC is defined as

$$\text{AIC}_p = N \ln \left( \epsilon_{\text{MSE},p} \right) + 2p. \tag{24}$$

where $p$ is the number of coefficients, $N$ is the sample size and $\epsilon_{\text{MSE},p}$ is the mean square error (MSE) for the respective parameter set.

For all considered signals of damages $S_1$ and $S_2$, as well as the reference $S_0$, of the two gearboxes with synthetic damages, the number of required prediction coefficients was calculated based on the AIC. For the gearbox with damage to gear $z_1$ a number of prediction coefficients $p_{z_1} = 17$ and for the gearbox with damage to $z_3$ a value of $p_{z_3} = 54$ results.

## 5. Calculation of Digital Filter Limits Using Difference Spectra

To determine appropriate filter limits, the linear prediction spectra $X_{\text{LP},S_1}$ and $X_{\text{LP},S_2}$ for damage levels $S_1$ and $S_2$ are calculated. Furthermore, $X_{\text{LP},S_0}$ describes the reference spectrum of the time windows $S_0$. First, difference spectra $\widetilde{X}_{\Delta,S_1} = X_{\text{LP},S_1} - X_{\text{LP},S_0}$ and $\widetilde{X}_{\Delta,S_2} = X_{\text{LP},S_2} - X_{\text{LP},S_0}$ are defined to determine the differing frequency ranges. Of particular interest here are the ranges where the difference is greater than 0 dB, since a more distinctive behaviour of the vibration signal due to the damage is recognisable. The difference spectra are shown in Figure 14 above for the gearbox with damage to gear $z_1$ and below for $z_3$.

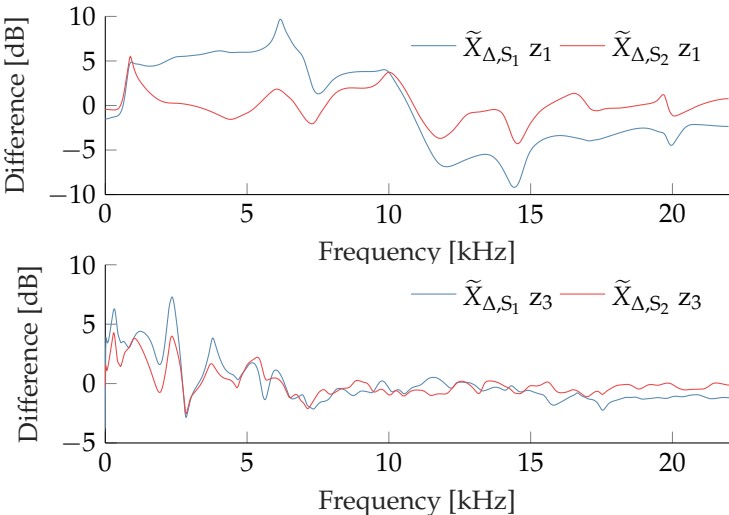

**Figure 14.** Comparison of the difference spectra $\widetilde{X}_{\Delta,S_1}$ and $\widetilde{X}_{\Delta,S_2}$ for both measurement series.

It can be seen that in both cases the largest damage $S_1$ shows the most distinctive behaviour. Overall, both difference spectra are strongly correlated. For the measurement series with damage to gear $z_1$ in the upper curve, it can be seen that the distinctive areas occur mainly in the lower frequency range, while damage $S_2$ shows a clearly narrower behaviour than $S_1$ in the frequency domain. For the measurement series with damage to $z_3$, a clearly more distinctive behaviour occurs in the lower frequency range. The differences between the frequency ranges are mainly associated with the different geometry of the gears and the different transmission paths of the vibration to the sensor. In order to determine appropriate digital filter limits for each gearbox stage, the frequency ranges with more distinctive behaviour are searched for in the following. For this purpose, the function $f^+(x)$ is defined, which limits the difference spectra to the positive range according to

$$f^+(x) = \begin{cases} 1, & \text{for } x > 0 \\ 0, & \text{for } x \le 0 \end{cases}. \tag{25}$$

This replaces negative function values to zero and positive to 1. The overlap regions $\widetilde{X}_{\text{overlap}}$ of the difference spectra are subsequently defined as

$$\widetilde{X}_{\text{overlap}} = f^+(\widetilde{X}_{\Delta,S_1}) \cdot f^+(\widetilde{X}_{\Delta,S_2}). \tag{26}$$

$\widetilde{X}_{\text{overlap}}$ thus describes equal distinct frequencies for both damage levels. Figure 15 shows the overlap of the difference spectra for the measurement series with damage at gear $z_1$ (blue) and gear $z_3$ (red). Frequencies greater than 11 kHz are not shown because no overlap occurs.

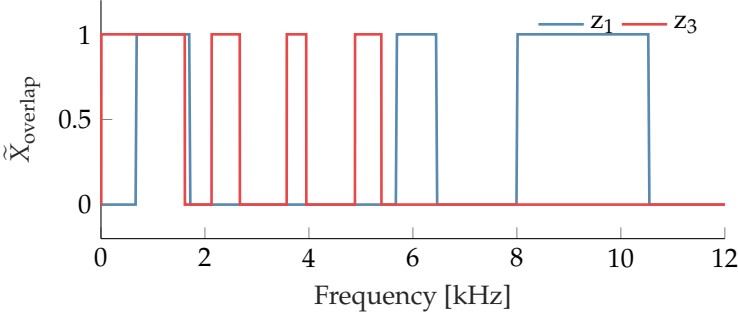

**Figure 15.** Overlapping frequency ranges of spectra $\widetilde{X}_{\Delta,S_1}$ and $\widetilde{X}_{\Delta,S_2}$.

The figure shows the frequency ranges of the distinct frequencies which occur in the difference spectrum of damage $S_1$ as well as damage $S_2$ for both examined gearbox stages. For the first gearbox with damage to gear $z_1$, frequencies from 687 Hz to 1697 Hz, 5693 Hz to 6445 Hz and 8013 Hz to 10,527 Hz result. For the gearbox with damage to gear $z_3$, frequencies from 0 Hz to 1611 Hz, 2129 Hz to 2669 Hz, 3574 Hz to 3947 Hz and 4887 Hz to 5392 Hz result. In the following, the determined frequency bands are examined exemplarily as filter limits for the examined measurements. For this purpose, a multiband filter with finite impulse response (FIR) is used. The results of the digital filtering are shown for the two investigated gearboxes in Figure 16. Both measurements were recorded at a rotational speed of 200 rpm and a torque of 30 Nm on the input shaft. Furthermore, the areas of the damaged tooth meshings $S_1$ (light gray) and $S_2$ (dark gray) are highlighted.

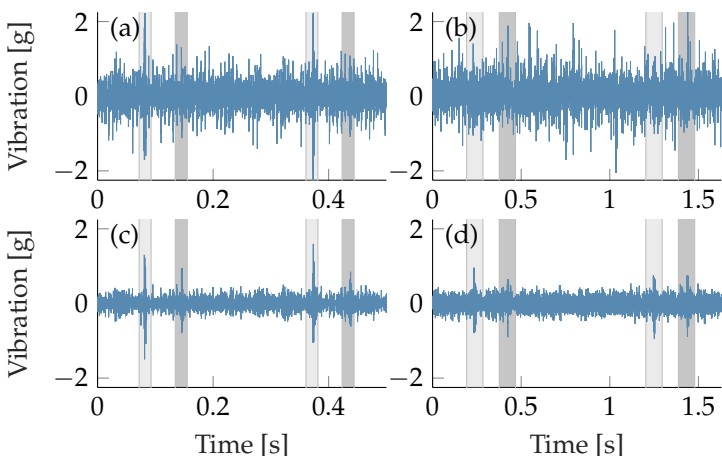

**Figure 16.** Results of multiband FIR filtering for the gearbox with damage on gear $z_1$ (**a**) unfiltered—(**c**) filtered and for the gearbox with damage on gear $z_3$ (**b**) unfiltered—(**d**) filtered. The rotational speed is 200 rpm and the torque is 30 Nm.

When looking at the unfiltered vibration signal, damage level $S_1$ can already be recognised in curve (a). Further damage cannot be seen directly in both (a) and (b). After digital filtering in (c) and (d), damage levels $S_1$ and $S_2$ can be clearly recognised in both cases. Furthermore, the influence of the other noises due to intact tooth meshing is considerably reduced. In addition, it can be seen that no further damage levels are recognisable in the vibration signal apart from damage levels $S_1$ and $S_2$. For both cases, two gear revolutions are shown and consequently damages $S_1$ and $S_2$ are seen twice each. For the calculation of the digital filter limits, torques between 20 and 42 Nm were considered. In the following, an additional example will be considered with two measurement series with torques of 70 Nm (at 220 rpm). This is shown with and without filtering as well as the highlighted damaged tooth meshes $S_1$ and $S_2$ in Figure 17.

From the figure it can be seen that damage $S_1$ and $S_2$ can clearly be recognised after filtering. Furthermore, it can be seen that the amplitudes of the signals have higher values due to the increased torque. Based on the results of the example, it is shown that measurement series with higher torque than used in the calculation of the filter limits can also be filtered successfully.

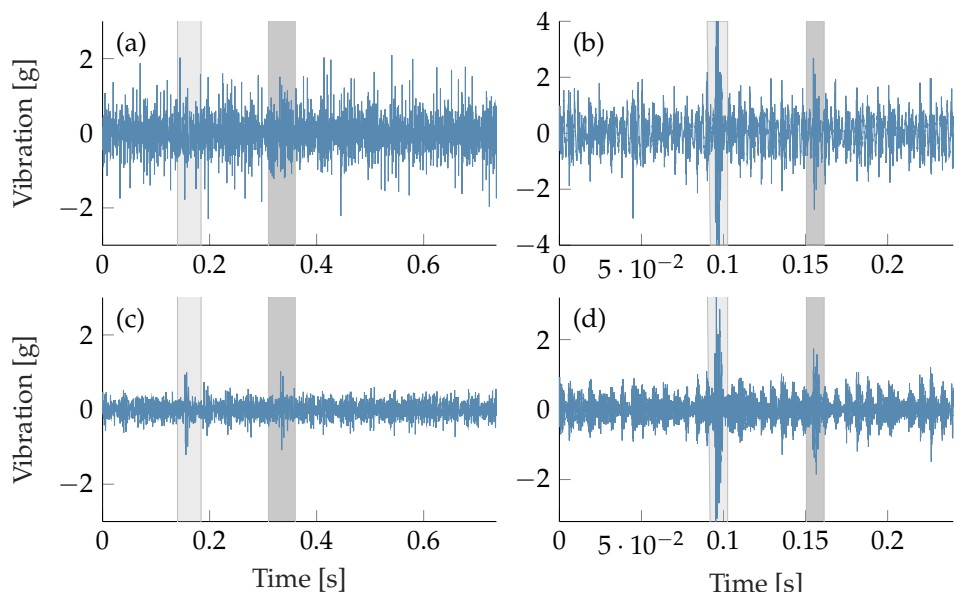

**Figure 17.** Results of multiband FIR filtering for the gearbox with damage on gear $z_1$ (**a**) unfiltered—(**c**) filtered and for the gearbox with damage on gear $z_3$ (**b**) unfiltered—(**d**) filtered. The rotational speed is 220 rpm and the torque is 70 Nm.

## 6. Conclusions

In this paper we proposed an investigation into the recognisability of gear damage levels in vibration signals. The recognisability of damage levels was investigated in the vibration behaviour of two different gears. We showed that on the first gearbox stage, damage levels up to a single pitch deviation of 73.48 µm could be recognised, and on the second gearbox stage damage levels up to 847.40 µm could be recognised. This implies that detectable size is gear-dependent. The vibration signals were investigated in the time domain using cross-correlation and on the basis of the signal components occurring in the frequency domain. In addition, frequency bands were determined for the recognisable damage levels in order to estimate appropriate digital filter limits. For this, linear prediction coefficients were used to calculate smoothed frequency spectra. Afterwards, the digital filter limits were determined using difference spectra. By applying the filter limits in a multiband FIR filter, it was finally shown that the digital filters make it considerably easier to recognise the damage levels in the time domain of the vibration signal. For future work, we plan to transfer the method to gearboxes installed in vehicles. In this way, the approach will be validated for environments with more disturbing noise. In addition, the investigated method is to be validated for the analysis of wear phenomena. Since the presented methodology finds the different frequencies between intact and damaged teeth, we assume a good transferability to wear phenomena.

**Author Contributions:** Conceptualization, A.B.; methodology, A.B.; software, A.B.; validation, A.B.; formal analysis, A.B.; investigation, A.B.; resources, A.B. and K.-L.K.; data curation, A.B.; writing—original draft preparation, A.B.; writing—review and editing, A.B. and K.-L.K.; visualization, A.B.; supervision, K.-L.K.; project administration, K.-L.K.; funding acquisition, K.-L.K. All authors have read and agreed to the published version of the manuscript.

**Funding:** This work is funded by the German Federal Ministry of Education and Research project VIPER (16ES0740).

**Data Availability Statement:** The data presented in this study are available on request from the corresponding author.

**Conflicts of Interest:** The authors declare no conflict of interest.

## Abbreviations

The following abbreviations are used in this manuscript:

| | |
|---|---|
| ACF | Autocorrelation function |
| AIC | Akaike information criterion |
| CCF | Cross-correlation function |
| DFT | Discrete Fourier transformation |
| DIN | Deutsche Industrienormen |
| FFT | Fast Fourier transform |
| FIR | Finite impulse response |
| LPC | Linear prediction coefficients |
| MSE | Mean square error |
| STFT | Short-time Fourier transform |

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
