# Peer review of "Investigations into the Recognisability of Gear Damage Sizes in Vibration Signals and Calculation of Appropriate Digital Filter Limits"

_applsci, doi:10.3390/app12094216_

Round 1
Reviewer 1 Report
The comments are as follows:
- More detailed gear parameters should be given, such as material, module, pressure angle, helix angle (spur or helical gears), accuracy grade (ISO/DIN), modifications, shift-coefficient, backlash, etc.
- What is the contact ratio of the two-staged gear box? The contact ratio significantly affects the vibrational level and signals but was not considered.
- The manufacturing errors, including involute profile deviation, single pitch deviation, adjacent pitch deviation, and total pitch deviation, already exist on the gears and the values of these factors depend on the accuracy grade of the gear. The tactile measurement results should reveal these deviations. These factors are not considered.
- 4: It is said the single pitch deviation is described in radian, but the unit in Table 1 is micron-meter?
- Four speeds were used for testing, and the speeds only extend from a low-level speed range, 100~400 rpm. Is this practical? How about higher speed conditions and nonlinear phenomena of gear dynamics?
- Applied torque was not described? The torque and the loaded transmission errors resulted from loading significantly affect the vibrational signals.
- The authors need to clarify whether the presented findings is only an observation from the limited cases or are applicable to detect gear tooth surface damage levels? This is related to the contribution level of this study.
Reviewer 3 Report
Congratulations to the authors for the work done, an interesting and well written work. However, some improvements or clarifications are required. See the comments below:
In the introduction the scientific contribution is not clear enough. It is clearly mentioned which analyses are presented, but it should be explained better what does this work prove or show respect to the current state of the art.
Resampling the signals from 51.2 kHz to 44 kHz does not introduce noise? Why not acquiring the signal at a 44 kHz sampling frequency?
How are the damaged tooth selected? How does the sequence of damaged and no damaged tooth affect the results?
For the spectrogram calculation why is a FFT length of 512 chosen? And how are the regions S1, S2, S3 and S4 identified?
In section 3.3 some statements should be corrected (instead of the words in bold should be that in green?):
- A window function is used to avoid aliasing in the
frequency signal. leakage - but the spectrum is smoothed by the increased 244
number of interpolation points. frequency resolution
Figure 11, Figure 12 and Figure 13 would be clearer if markers were used, or different line style.
The frequency range where differences are noticeable is observe in Figure 14. It is different for z1 and z3, any explanation can be given regarding the frequency range to take into account? How is it related to rotational speed and damaged tooth?
Figure 15 should be referenced in the text.
Equation 25 should be corrected. für is written.
Finally, could the authors add some future lines to the conclusions?
Round 2
Reviewer 1 Report
The authors have replied the reviewer's comments and made necessary revision on this manuscript. There is only one point to remind that the gears accuracy ISO grades of 9 and 11 are not high accuracy grades. Practically high accuracy grades are ISO 4. The description on page 4 below Table 1 should be revised.
Author Response
Dear reviewer, first of all we would like to thank you again for your comment and time spent to improve our contribution!
Your comment pointed out an issue that we have revised. Thank you for this comment!